# Detection of Physiological Signals Based on Graphene Using a Simple and Low-Cost Method

**DOI:** 10.3390/s19071656

**Published:** 2019-04-06

**Authors:** Liping Xie, Xingyu Zi, Qingshi Meng, Zhiwen Liu, Lisheng Xu

**Affiliations:** 1School of Sino-Dutch Biomedical and Information Engineering, Northeastern University, Shenyang 110819, China; xielp@bmie.neu.edu.cn (L.X.); zi_xingyu@163.com (X.Z.); 2Aerospace Engineering, Shenyang Aerospace University, Shenyang 110136, China; mengqingshi@hotmail.com (Q.M.); zhiwenliu86@163.com (Z.L.); 3Neusoft Research of Intelligent Healthcare Technology, Co. Ltd., Shenyang 110167, China

**Keywords:** graphene, thermal expansion, resistance, sensors, physiological signals

## Abstract

Despite that graphene has been extensively used in flexible wearable sensors, it remains an unmet need to fabricate a graphene-based sensor by a simple and low-cost method. Here, graphene nanoplatelets (GNPs) are prepared by thermal expansion method, and a sensor is fabricated by sealing of a graphene sheet with polyurethane (PU) medical film. Compared with other graphene-based sensors, it greatly simplifies the fabrication process and enables the effective measurement of signals. The resistance of graphene sheet changes linearly with the deformation of the graphene sensor, which lays a solid foundation for the detection of physiological signals. A signal processing circuit is developed to output the physiological signals in the form of electrical signals. The sensor was used to measure finger bending motion signals, respiration signals and pulse wave signals. All the results demonstrate that the graphene sensor fabricated by the simple and low-cost method is a promising platform for physiological signal measurement.

## 1. Introduction

There has been a growing demand for wearable flexible sensors with the continuous advance of modern technology. Flexible and wearable sensors have drawn extensive attention due to their wide potential applications in wearable electronics. Flexible sensors with high sensitivity, good flexibility, and excellent stability are highly desirable for monitoring human biomedical signals, movements and environmental factors [1,2]. To ensure a robust and conformal contact with the curvilinear, rough, and dynamic surface of the skin without impeding daily activities, wearable sensors should have a low modulus and high stretchability. In addition to good wearability, wearable sensors with high sensitivity, light weight, low cost, low power consumption and negligible hysteresis are desirable [3,4,5]. Nanomaterials possess larger surface areas, exceptional material properties and are compatible with low-cost fabrication processes. Thus, nanomaterials are widely employed as building blocks for developing wearable sensors [6]. Flexible sensors based on nanomaterials have opened a new door for the accurate and comfortable measurements of physiological signals [7,8,9]. The ideal graphene is monoatomic layered graphite with a two-dimensional honeycomb crystal structure formed by SP^2^ hybridization [10,11]. Its basic structure is composed of six-membered ring-like benzene units, and it is one atom thick. Due to the special structure of graphene, it has many unique properties, excellent electrical properties (electron mobility up to 2 × 105 cm^2^/(V·s)), extraordinary thermal conductivity (5000 W/(m·K)), outstanding specific surface area (2630 m^2^/g), excellent Young’s modulus (1100 GPa) and tensile strength (125 GPa) [12,13,14]. Graphene plays an important role in the field of wearable sensors due to its sensitive piezoresistive effect, large electrical conductivity and excellent Young’s modulus [15,16,17]. Graphene-based materials have shown the highest gauge factors among all reported literature and can be transferred to various flexible substrates [18]. It is possible to exploit piezoresistivity variations in order to measure pressure [19,20].

A variety of graphene piezoresistive sensors have been developed for detecting human motions and spatially resolved monitoring of pressure distribution [21,22,23]. Graphene-based sensors high sensitivity, and are wearable and flexible for health monitoring [24,25,26,27]. Sang-Hoon Bae et al. used the method of chemical vapor deposition (CVD) to prepare graphene, and chose polydimethylsiloxane as a flexible substrate. Finger bending signals were measured [28]. They proved the feasibility of graphene as a conductive material in a flexible strain sensor. However, the CVD method significantly increases the fabrication cost. Graphene strain sensors can be fabricated by laser engraving technology with different patterns [29]. However, the methods of production and patterning are complex. The patterning requires at least 25 min, which is inefficient and time-consuming for mass production. Wang et al. developed graphene woven fabrics (GWFs), which are fabricated on a crossover copper mesh by CVD. The GWFs can be used to sensitively measure human body motion signals [30]. Unfortunately, the method is time-consuming, and has the disadvantages of photoresist contamination and the cracking graphene films. Electrostatic textile technology and redox method have been used to produce graphene pressure sensors by wrapping reduced graphene oxide (rGO) on electrospun poly vinylidenefluoride-co-trifluoroethylene (PVDF-TrFE) nanofibers. The graphene pressure sensors can monitor human body signals in real time [31], but this requires tedious processing steps and a long production time (over 20 h). The preparation of a graphene pressure sensor for physiological signal detection using a simple method is still in its infancy.

In this paper, a graphene sensor is demonstrated for physiological signal measurement. Graphene nanoplatelets (GNPs) are synthesized by thermal expansion method, and the graphene sensor is fabricated by double-sided sealing of polyurethane (PU) medical film on the graphene sheet. Three kinds of physiological signal are measured, including finger bending signals, respiratory signals and wrist pulse signals. A signal processing circuit composed of a signal conversion circuit and a filter circuit is developed to acquire and transmit the physiological signals. The GNPs synthesized by thermal expansion are cheap, conductive and already commercially available compared with CVD graphene and liquid phase exfoliated graphene [24]. Due to its ease of production, the versatility of the technique, the low cost and the potentially large scalability, the as-synthesized sensor is believed to be promising in applications of rehabilitation training, sleep monitoring and heart disease prevention.

## 2. Experimental Materials and Experimental Methods

### 2.1. Experimental Materials and Instruments

The graphite intercalation compound was bought from Asbury Graphite Mills Inc (Asbury, OH, USA). PU films were bought from Shanghai Hons Medical Technology Co. Ltd. (Shanghai, China. The graphene sheets were formed by a manual digital display pressure machine YP-12BS (Tianjin, China). The tensile measuring of the GNP sheet was performed on a tensile measure machine TST-01 (Jinan, China). Scanning Electron Microscope (SEM) images of the GNP sheets were obtained by JEOL JSM-7800FPRIME (Tokyo, Japan). The X-ray diffraction (XRD) spectra of graphite and GNPs were characterized by X-ray polycrystalline diffractometer APD 2000 PRO (Beijing, China). The resistances were recorded by a Hydra Series III Data Acquisition System Fluke 2638A (Everett, USA). The output voltages of the signal processing circuit were recorded by a data acquisition instrument NI USB-63619 (Beijing, China).

### 2.2. Experimental Methods

GNPs were prepared by thermal expansion and exfoliation method [31]. The thermal expansion and exfoliation method peeled flake graphite at high temperature. First, a small amount of commercial graphite intercalation compound (Appendix A) was put in a crucible which had been preheated to 700 °C. Then, the graphite intercalation compound in muffle furnace was maintained at 700 °C for one minute. The thermal expansion converted the compounds into worm-like materials (Appendix A), producing a large amount of GNPs. Due to the low density of GNPs, precautions must be taken in the process to avoid inhaling GNPs into the lungs. 0.04 mg of the synthesized GNPs was weighed. Then, the synthesized GNPs were placed in a mold to make a graphene sheet by an electronic pressing machine. The mold is 50 μm high, 20 mm long and 10 mm wide. It is worth noting that when the GNPs were placed in the mold, a slight pressure was applied to them in order to prevent the diffusion of GNPs at the edge of the mold and the uneven thickness of the graphene sheet. Then, a graphene sheet with a thickness of 48 μm ± 5 μm was formed by keeping the pressure at 1 MPa for 1 min in a manual digital display pressure machine (Appendix A). Two pieces of medical PU film were used to packet the graphene sheet to form a graphene sensor. 

## 3. Results and Discussion

### 3.1. Characterization of the Graphene Sensor

As shown in Figure 1, we measure the X-ray diffraction (XRD) spectra of graphite and GNPs by X-ray polycrystalline diffractometer. It shows that the diffraction peak of graphite is very sharp at a theta of 26.5°. The corresponding layer spacing is 0.336 nm by calculating the Prague equation, which indicates that the degree of graphitization is high, and the spatial arrangement of the microcrystalline layer is highly regular. The diffraction peak of the GNPs is still 26.5°; however, the intensity of the diffraction peak is much lower, and the half-width of the diffraction peak is wider compared with the graphite, which demonstrates that the graphite with the complete crystal structure is transformed into the GNPs with the decrease of integrity and the increase of disorder degree. The results show that amorphous GNPs has been successfully obtained [32]. Figure 2 shows the frontal and side cross-section views of the GNP sheet. The smooth and continuous surface forms a conductive network-layer. The side cross-section of the GNP sheet confirms that the GNP sheet consists multilayers of conductive networks. The conductive multilayer structure enables stretchability of the GNP sheet, and provides a continuous conductive network for sensing.

To verify the feasibility of the GNP sheets as a sensor, we characterized the pressure response of the GNP sheets. A GNP sheet is placed on the pressure measuring machine. Different resistances of the GNP sheet corresponding to different pressures are recorded. Figure 3a shows the relative resistance variations of the GNP sheets under different pressures. Applying linear fitting to the data, we get the best fitting lines with R^2^ of 0.98 at pressure ranging from 0 to 20 KPa and R^2^ of 0.99 at pressure ranging from 20 to 80 KPa. The changes of relative resistance variations of the GNP sheets have a bilinear relationship with the variation of the pressure applied to the sheet. The pressure response of the GNP sheets verifies that the GNP sheets have a high relative sensitivity at low pressures (0 to 20 KPa), which is suitable for practical applications (e.g., gentle touch, figure bending, object manipulation). In addition, the piezoresistive properties of the GNP sheets are measured and analyzed. A GNP sheet connected with conductive wires was fixed at both ends on the tensile measuring machine. The GNP sheet was stretched slowly. In the meantime, its resistances were recorded at different stretching lengths. The experimental results were calculated using 3 repeated sheets. The initial resistance R_0_ of the GNP sheet is 6.5 Ω. Figure 3b shows the relative resistance variation of the GNP sheet versus stretching strain. Having employed a linear fitting to the data, we get the best fitting line with R^2^ of 0.98. This fitting line shows that the change of relative resistance variation of the GNP sheet has a linear relationship with the variation of the stretching strain of the GNP sheet, which provides a good foundation for measuring physiological signals based on the graphene sensor. The GNP sheet is relatively flexible due to its condensed multiple-layer structure. It can bend to almost 270 degrees (Appendix A). When the GNP sheet is coupled with PU film, the mechanical properties are further enhanced by the PU film. The response of the graphene sensor to strains is presented in Figure 3c. The graphene sensor responds in a good linear relationship to the applied strain, with the strain load during the experiment ranging from 0% to 55%. The deformation of the GNP sheets can only be stretched to 13%, and will break upon further stretching. On the contrary, the GNP sheets packed with PU film can be stretched up to 100% without mechanical failure, resulting in a significantly improved stretchability after PU film sealing. The gauge factor k is ∼205.54 for the GNP sheets in the deformation range from 0% to 55% strain, and ∼58.83 for the graphene sensor in the deformation range from 0% to 10% strain. The gauge factors for our graphene sensor (achieved ∼58.83 at 55% strain) are comparable with previous graphene-based strain sensors [21,22,23,33]. The hysteresis behavior of the sensors is also shown in Figure 3d. In the stretching process, the value of ΔR/R_0_ steadily increases upon stretching, and gradually decreases during the release process. Hysteresis is mainly caused by the viscoelastic nature of polymers, as well as the interaction between nanomaterial fillers and polymers [3]. Hysteresis occurred in the graphene sensor, since PU film is a viscoelastic polymer matrix with elasticity and viscosity.

### 3.2. Fabrication of Signal Processing Circuit

The graphene sensor is used to detect physiological signals, including finger bending signals, respiration signals and wrist pulse signals. The overall schematic of measuring physiological signals is shown in Figure 4a. First, the physiological signals of the human body are detected by the graphene sensor, and the conversion of the graphene sensor resistance signal to the voltage signal is realized through a signal conversion circuit. A Wheatstone bridge circuit and an operation amplifier (AD620) are composed of the signal conversion circuit (Figure 4b). The AD620 gains resistor R6 is set at 10 KΩ, and the gain formula is shown in Equation (1),
(1)G=49.4KΩRG+1

The measured resistance R_x_ in the bridge is the resistance of the graphene sensor. The reference resistance R_3_ is chosen as 120 Ω. Then, the output voltage of the signal conversion circuit theoretically follows the following Equation (2),
(2)V=5(Rx( Rx+R5)−0.5)(49.4R6+1)

The linear range of the output voltage of the signal conversion circuit is between −2.4 V and 3.6 V due to the AD620 accessing a direct current (DC) power of 5 V. In the experiment of physiological signal measurements, the output of the bridge circuit conforms to the linear range when the graphene sensor is placed in the bridge circuit, which benefits the physiological signals measurements without signal distortion. 

To improve the signal-to-noise ratio and reduce the noises produced by external noise and internal power frequency noise, filter circuits with different filtering ranges for measuring different signals are fabricated. As shown in Figure 4c, the filter circuit consists of a two-order active high-pass filter circuit and a two-order active low-pass filter circuit, which can be referred to as a two-order active bandpass filter circuit. R_7_, R_8_, R_11_, R_12_, C_1_, C_2_, C_3_ and C_4_ are parameters of the filter circuit and can be changed according to the required filter frequency range of the detection signal. The high cut-off frequency of the filter circuit can be expressed as in Equation (3),
(3)fH=12πC1C2R7R8

The low cut-off frequency is expressed as in Equation (4),
(4)fL=12πC3C4R11R12

The filter circuit not only possesses a filtering function, but also an amplification function. The amplification factor can be expressed as in Equation (5),
(5)A0=1+R9R10

The signal conversion circuit converts the resistance signal acquired by the graphene sensor to a voltage signal. The filter circuit removes the noise from the signals. The whole system makes it possible to improve the signal-to-noise ratio and amplify the voltage amplitude of the signal by 11.4 times.

### 3.3. Measurement of Finger Bending Action

Finger bending action measurement is of great importance in the field of human health and robotics [34]. The prevention, treatment and rehabilitation of cardiovascular and cerebrovascular diseases have gradually attracted people’s attention because of the increasing incidence of cardiovascular and cerebrovascular diseases. The sequelae of cardiovascular and cerebrovascular diseases are often manifested in language, behavior disorder and memory decline. Behavior disorders are particularly manifested in the disorders of finger control. The best period of rehabilitation treatment is within two months after clinical treatment, and the most effective rehabilitation therapy is needed within this time range, otherwise, the sequelae will have a great impact on the life and work of the patients. Hence, timely rehabilitation therapy is of great significance to patients. At the same time, the effect of rehabilitation is reflected in human movement signals during rehabilitation. It is important to monitor the physical signal during rehabilitation training. 

We apply the graphene sensor to monitor finger bending signals in real time. We choose the low-pass filter circuit at 10 Hz to filter the finger bending signal, and collect the signal with a sampling frequency of 10,000 Hz using the data acquisition instrument (Video). Figure 5 shows the signals of the finger bending action. State A is a resting state before flexion of the fingers, and state B corresponds to the greatest degree of finger bending during the measurement. The voltage values of the signals between state A and state B decrease first, and then increase compared with state A, indicating that the resistance values of graphene sensor decrease firstly and then increase. Both pressure and tension contribute to the changes of resistance values from state A to state B. The graphene sensor is subjected to pressure firstly in this process, so that the resistance values decrease, and then the resistance values increase because the tension plays a major role. State C is the state from the bending to the resting state. When the finger moves back to its original position, the voltage values of the signals decline firstly, and then increase. This is because the stretching strain of graphene sensor firstly decreases, so that the voltage values of the graphene sensor decrease. In addition, then the pressure dominates, which causes the voltage values to fall below zero. With the release of pressure and tension, the voltage of the graphene sensors returns to the original state. The data show that the collected voltage signal can reflect the finger bending action. By extracting and calculating the key points of the figure bending signals with MATLAB, we find that if the finger moves periodically, the system outputs a periodic voltage signal. The results demonstrate that the graphene sensor can monitor the movement of fingers, which can be used in the rehabilitation of finger flexion. It provides an intuitional method to observe the effect of rehabilitation training, which is important for rehabilitation training of cardiovascular and cerebrovascular diseases.

### 3.4. Measurement of Respiration Signals

Respiration is an important physiological process of the human body. Detection of respiration signals is an important part of healthcare monitoring. Diseases related respiration signals have attracted wide attention all over the world. The incidence of obstructive sleep apnea syndrome (OSAS) is high and harmful. Accurate and reliable monitoring of patients’ respiratory and sleep physiological signals can help to cure disease [35]. At present, sleep monitoring devices are expensive and are limited to specific application scenarios. Therefore, the development of a portable and low-cost respiratory sleep monitoring system is of great significance. The fabricated graphene sensor was attached to the abdomen of a volunteer by a PU film. The recorded signals were recorded and processed by the signal processing circuit. The output signals were obtained by a data acquisition instrument. A low-pass filter circuit at 20 Hz and a sampling frequency at 10,000 Hz are chosen to filter the respiration signals and collect the signals by data acquisition instrument. The collected signals are analyzed by MATLAB. The three points A, B and C in Figure 6 are three moments of the respiratory process. Moment A is the beginning of inbreathing. The pressure in the lung begins to increase. Moment B represents a moment of starting expiration, corresponding to the beginning of the pressure decreasing in the lung. Moment C is a moment of finishing expiration. This is also the beginning of a resting state. As with the mechanism analyzed in the finger bending signal, both pressure and tension contribute to the changes in voltage values from moment A to moment C. After the extraction and calculation of the feature points, the deep breathing cycle of the subject is 5.25 s, with a frequency of 11.42 times per minute. The respiratory rate accords with the depth of respiratory frequency. It shows that the graphene sensor has good accuracy in measuring respiratory signals. This graphene sensor can achieve effective and low-cost detection of respiration signals during sleep, which is of great significance for the prevention and treatment of respiratory diseases.

### 3.5. Wrist Pulse Signal Measurement

Pulse signal is an auxiliary examination of cardiovascular diseases. High pulse rate has a high correlation with the incidence of coronary heart diseases [36]. It can predict hypertension, coronary heart disease and other diseases. When the pulse rate of an adult is more than 100 beats per minute, it is known as tachycardia. If the pulse rate of an adult is less than 60 beats per minute, it is called bradycardia. There are many diseases in clinic will result in the change of the pulse rate. For example, the speed of heart will be accelerated when fever occurs, resulting in an increase in the pulse rate, and especially the heart disease can change the pulse rate. The measurement of pulse rate has become an indispensable testing item for patients. 

The graphene sensor is applied to measure wrist pulse signals. We attach the graphene sensor to the wrist to measure the radial artery pulse wave. The pulse wave pressure on the wrist skin surface is less than 0.6 KPa [37,38], which conforms to the linear range of the resistance of the graphene sensor varying with the pressure applied to it. The pulse signals are filtered using the bandpass filter circuit with 0.5–25 Hz. A remarkable periodic voltage signal is obtained, as shown in Figure 7a. The peaks of the pulse signal are marked with red circles. To check the quality of the acquired pulse signal, we analyze the pulse wave signal by spectrum analysis (Figure 7b), and compare it with the normal pulse wave signal spectrum obtained using traditional sensors (Figure 7c). From the comparison between Figure 7b and Figure 7c, we draw the conclusion that the pulse signal measured by the graphene sensor accords with the main frequency distribution range of the normal pulse wave. After calculating the interval between one peak and its adjacent peak (PP) of the pulse signal, we find that the average PP interval is 0.79 s, and the variance is 0.03. The results demonstrate that the wrist pulse of the subject is in accordance with the normal pulse wave signal with a range of 0.75 s–0.85 s. All these results show that the graphene sensor provides a low-cost, simple to produce, high-comfort and accurate measurement method for the pulse rate, which is attractive in the field of preventing hypertension, coronary heart disease, and other diseases.

## 4. Conclusions

In short, we synthesized GNPs using the thermal expansion and exfoliation method, and developed a graphene sensor. The resistance of the graphene sensor has a linear relationship with its stretching strain, which lays a solid foundation for the measurement of physical signals. We fabricated a signal processing circuit to output the physiological signals collected by the graphene sensor. The data of the finger bending signal, respiration signal and wrist pulse signal were processed and analyzed, which confirmed the accuracy of the signals. The graphene sensor fabricated by this method provides a low-cost, simple and flexible sensing method for measurement of physical signals. This is believed to be promising in applications of rehabilitation training, sleep monitoring, and heart disease prevention. 

## Figures and Tables

**Figure 1 sensors-19-01656-f001:**
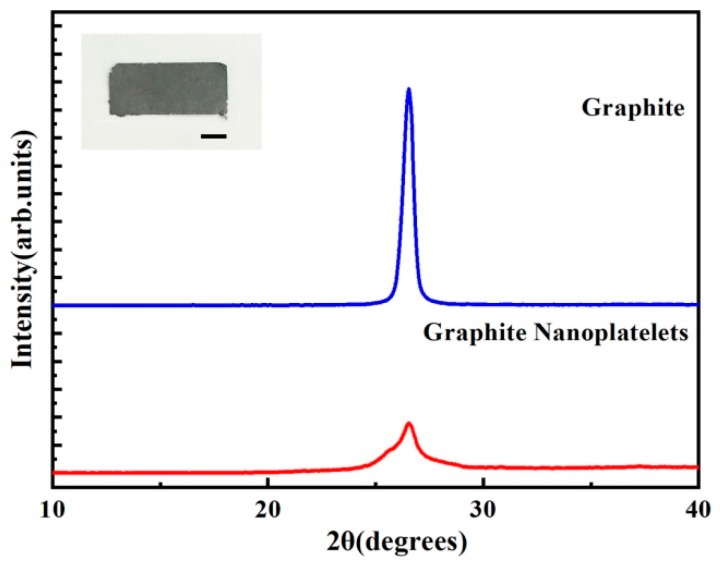
XRD diffraction patterns of graphite and GNPs. The inserted picture is the photograph of GNP sheets. The scale bar is 5 mm.

**Figure 2 sensors-19-01656-f002:**
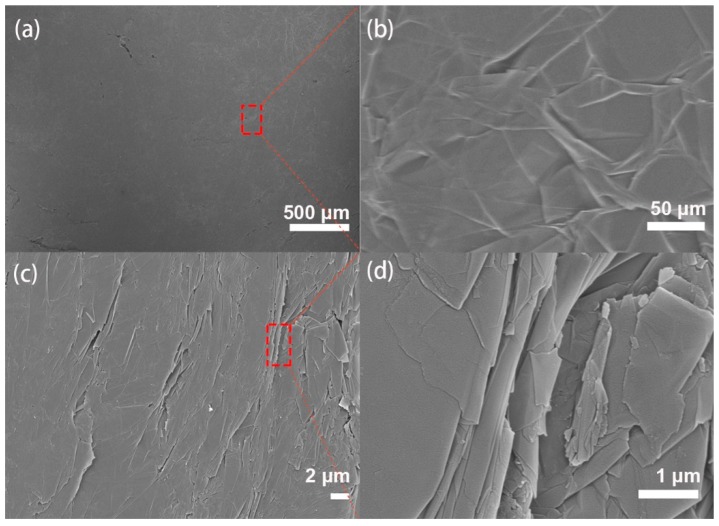
SEM images of GNP sheets. The morphology of the frontal (**a**,**b**) and side cross-section (**c**,**d**).

**Figure 3 sensors-19-01656-f003:**
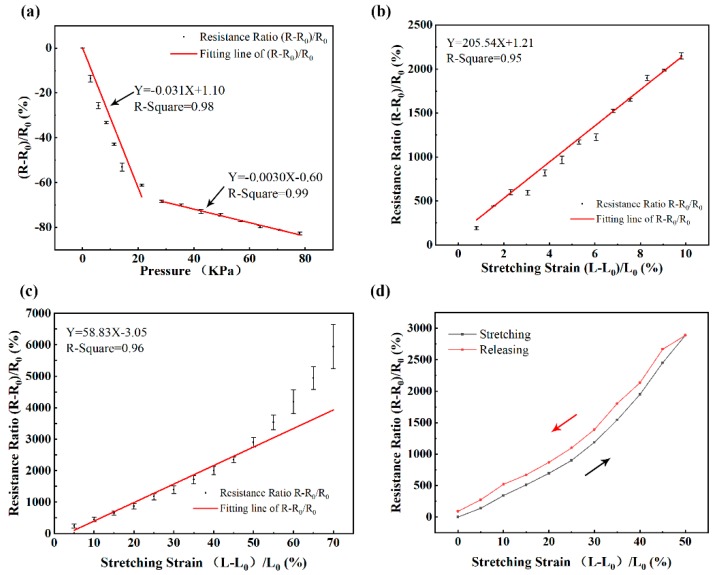
The mechanical response of the GNP sheets and the graphene sensors. (**a**) The relationship of the resistance ratio of the GNP sheet with the applied pressure. (**b**) The relationship of the resistance ratio of the GNP sheet with its stretching strain. (**c**) The relationship of the resistance ratio of the GNP sheet packed with PU film with its stretching strain. (**d**) Hysteresis performance of the graphene sensor. R_0_ is the initial resistance value. R is an instantaneous resistance value. L_0_ is the length of the GNP sheet. L is the instantaneous length of the GNP sheet.

**Figure 4 sensors-19-01656-f004:**
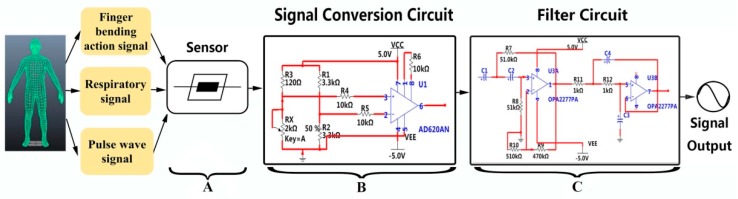
Schematic of sensing and measurement circuits. Part A is the graphene sensor. Part B is the signal conversion circuit. Part C is the filter circuit.

**Figure 5 sensors-19-01656-f005:**
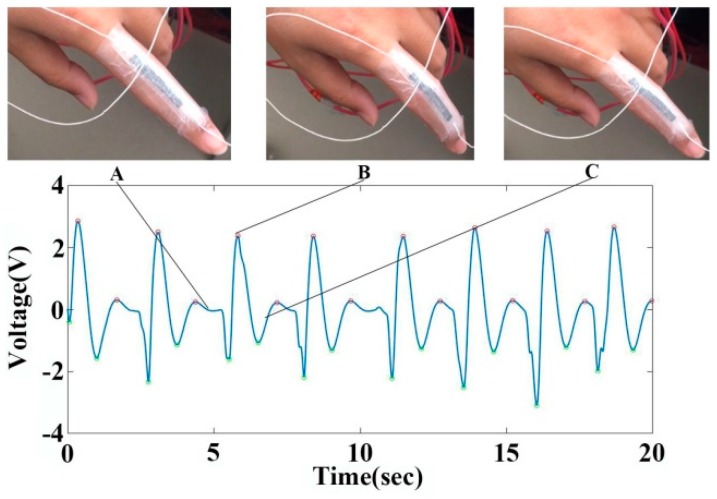
Signal acquired by graphene sensor for monitoring finger bending motion. A, B and C correspond to the finger resting state, finger flexion state and finger recovery state, respectively.

**Figure 6 sensors-19-01656-f006:**
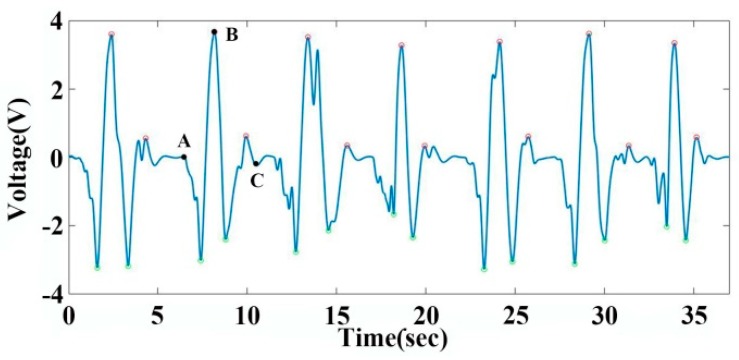
Signal acquired by graphene sensor for monitoring deep respiration, A, B and C corresponding to the beginning moment of inbreath, starting moment of expiration and beginning of a resting state, respectively.

**Figure 7 sensors-19-01656-f007:**
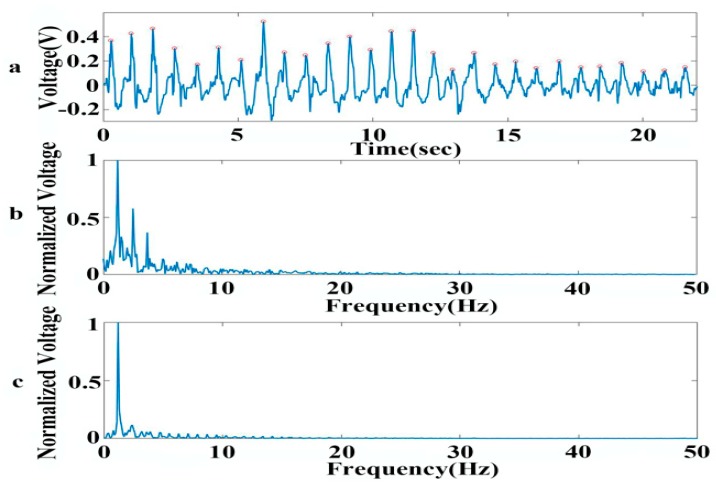
Signal acquired by graphene sensor for measuring pulse wave. (**a**) Voltage signal obtained by graphene sensor. (**b**) Spectral analysis of the pulse signal by the graphene sensor. (**c**) Spectral analysis of the pulse signal measured by a traditional sensor.

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
