# Peer review of "Detection of Physiological Signals Based on Graphene Using a Simple and Low-Cost Method"

_sensors, 2019, doi:10.3390/s19071656_

Round 1
Reviewer 1 Report
The manuscript “Detection of Physiological Signals Based on Graphene Using a Simple and Low-cost Method” report about an easy, versatile and potentially large-scalable way to produce sensors graphene-based to detect physiological signal. Although interesting, the paper needs major revision.
General Comments
Introduction
In general, the introduction of the paper should present and discuss also the positive achievement in the field of physiological signal measured trough graphene-based sensors. Then some example of complications (e.g. in the manufacturing) have also to be presented and the advantage of your technique should be clearly stated (for example easiness of production, versatility of the technique, potential large scalability). The authors should focus more in the presentation of positive achievement. Some of the references suggested in the specific comments could be used also to give examples of positive achievements in the field of physiological signal measured trough graphene-based sensors. The authors should also be more precise in presenting the advantage related to their work. Are there other works that couple polyurethane and graphene to make stretchable materials for electronics and, in particular, for electrophysiological signal detection and processing? The authors should discuss this more in detail and provide closest prior art (also related to this topic, they can find some useful reference in the ones indicated in the specific comments). Moreover, in the introduction the role of hysteresis for strain sensing should be mentioned and discussed.
Experimental materials and experimental methods
The authors should report in details how they measured the resistance variation with strain. They should also report here how was their setup to measure all the physiological signals. They should say on how many samples they performed the measurements.
Results and discussion
The authors should report the initial value of sheet resistance of the GnPs sheet. The authors should describe how mechanically is the graphene sheet (is it flexible, is it brittle?). The authors should report how these mechanical properties change when the GnPs sheet is coupled with polyurethane. The authors should discuss and possibly report data about hysteresis of their graphene-based sensor.
Specific Comments
- Line 30: reference needed. Suggested: https://doi.org/10.1002/adfm.201504755, https://doi.org/10.1002/advs.201700587, doi:10.1088/0957-4484/26/37/375501. Here the authors could also mention and discuss hysteresis.
- Line 32: What the authors mean saying that nanomaterials are compliant?
- Line 35: reference needed. Suggested: https://doi.org/10.1002/adma.201504150, https://doi.org/10.1002/aelm.201600245, https://doi.org/10.1016/j.molmed.2010.08.002.
- Lines 37: reference needed. Suggested: DOI: 10.1126/science.1102896, doi:10.1038/nature04233.
- Line 38: Graphene does not have a minimum thickness. Graphene is defined as one atom thick. Therefore, is not “comparable with one atom thickness” but it is one atom thick. Please correct. It is also worth to introduce more clearly the material that the authors are using (graphene nanoplatelets, see the following comments).
- Line 45: The authors need to be more specific. The authors should mention that for measuring pressure it is possible to exploit piezoresistivity or capacity variations. Please use appropriate references (e.g. https://doi.org/10.1016/j.mee.2015.06.007).
- Line 50: the authors say that CVD method increase the fabrication cost. This is true. The authors should introduce graphene nanoplatelets and say that they chose to use this material because it is cheap, conductive and already commercially available compared with cvd graphene and single layer graphene liquid phase exfoliated. The authors can use this reference (https://doi.org/10.3390/app8091438) to clarify this point.
- Lines 50-54: Most of the papers in literature that deal with pdms sensor graphene based does not need any laser processing (e.g. https://doi.org/10.1063/1.4826496, https://doi.org/10.1002/adfm.201602619, https://doi.org/10.3390/s16122148). The authors should discuss this more in details or remove the laser processing in the drawbacks of pdms sensors. Please use the previously mentioned references for this discussion.
- Lines 65-67: the authors should clearly indicate here which are the advantages of their work compared with previous works (e.g. low cost, versatility, easy manufacturing etc).
- Lines 79-83: do the authors have references in which they have published already such protocol? If not, could they provide SEM of the obtained sheet (morphology and cross section). How they measured the thickness of 50 micron? How much is the thickness after the coupling with polyurethane? Can they provide SEM cross sectional image of the GnPs sheet packed in between the polyurethane?
- Lines 101-102: it is not clear which is the starting value of sheet resistance of the GnPs sheets. The authors should report it.
- Figure 2: How are the mechanical properties of the GnPs sheet? Is it brittle? What happen to the GnPs sheet when it is stretched? Mechanical characteristics should be reported (for example elongation and stress at break). How the mechanical properties change when the polyurethane embed the GnPs?
- Lines 161-162: reference needed for human health and robotics.
Author Response
Thank you for your valuable comments. Attachment please find the response.

Reviewer 2 Report
This paper reports the preparation of graphene thin film sensors based on thermally expanded graphene nanoplatelets. The novelty of this work is not clearly addressed as compared to other graphene-based sensors. I recommend major revision before being considered for publication. My comments are given in the following:
1) A general revision of the English is warmly recommended. Examples include:
First page line 16: “lays a solid foundation” not “laies a solid foundation”.
First page line 13: “double-sided coating of polyurethane (PU) medical film on a graphene sheet” is misleading. The graphene sheet is sandwiched by PU film. It is not considered as “coating”. Speaking of PU film encapsulation, how good the bonding is between graphene and PU film? Will the graphene sheet debond from PU film when being used as a strain sensor? Any evidence?
Fourth page line 122: “R0 is a value of the initial resistance value.” changed to “R0 is the initial resistance value.”
Not all the mistakes are listed here.
2) It is worth comparing the sensing performance of the sensors, including the stretchability and the gauge factor with other reported graphene-based strain sensors. Examples include a) Ultrasensitive and Stretchable Strain Sensors Based on Maze-Like Vertical Graphene Network. ACS Applied Materials & Interfaces 2018, 10 (42), 36312-36322. b) Strain Sensors with Adjustable Sensitivity by Tailoring the Microstructure of Graphene Aerogel/PDMS Nanocomposites. ACS Applied Materials & Interfaces 2016, 8 (37), 24853-24861. c) Wearable and Highly Sensitive Graphene Strain Sensor for Precise Home-Based Pulse Wave Monitoring. ACS Sensors 2017, 2 (7), 967-974. d) Highly Elastic Graphene-Based Electronics Toward Electronic Skin. Advanced Functional Materials, 27, 1701513.
3) In Figure 2, is the x-axis strain or extension? Strain dose not have a unit of “mm”.
4) For the detection of finger joint bending, I have several questions:
a) The authors mention that State B corresponds o the greatest degree of finger bending. What is the bending degree? To me, based on the figure 5B, the bending degree is less than 45°. The finger can, in reality, be bent to more than 90°. Based on the Figure 2, the graphene sensors can only be stretched to 2% of strain. Bending finger joints can generate strain up to 45% (Adv. Mater. 2014, 26 (13), 2022-2027.) In this case, the sensors should have already failed at this strain.
b) How is the resistance changing during the bending of fingers. The authors should compare the data in Figure 5 with that in Figure 2.
c) The sensors show much higher sensitivity to strain than to pressure. So, how the pressure can dominate?
d) Is there any reason that the authors present data with voltage as the Y-axis in Figure 5-7? Why not relative resistance changes like that in Figure 2. This is easy to compare and know how much strain is generated during these body movements.
5) The details of how the sensors are attached during the physiological signal measurements should be given. In particular, it is not clear how the sensors are attached during respiration measurement.
Author Response

(The authors gave the same response as above.)

Round 2
Reviewer 1 Report
The manuscript "Detection of Physiological Signals Based on Graphene Using a Simple and Low-cost Method " is enhanced after the revisions. English can still be improved. I suggest its publication after minor revisions.
Comments and Suggestions
- Lines 42-43: the authors should define what they means with good piezoresistive effect. Graphene-based materials have shown gauge factors among the highest ever reported. The authors can use this reference: https://www.nature.com/articles/srep00870.
- Line 50: use the past tense when describing something that was already done. "They proved".
-Line 57: This sentence is not clear.
-line 135: the authors should report also the value of sheet resistance (Ohm/sq) of the material.
- When the authors discuss hysteresis (Figure 3d), it would be interesting to know how much larger is the value of the resistance after the release of the stress.
Author Response
Dear Reviewer,
We thank the referee for carefully reading this manuscript and providing useful comments and suggestions. We have modified this manuscript to accommodate all the comments as indicated below (comments are indicated in bold; our responses are provided in italics and blue). All the specific changes have been made in response to the points in the revised manuscript in red.
Comments: The manuscript "Detection of Physiological Signals Based on Graphene Using a Simple and Low-cost Method " is enhanced after the revisions. English can still be improved. I suggest its publication after minor revisions.
Answer: We gratefully thank the reviewer for the positive comments. We have carefully checked and revised the manuscript as the reviewer’s suggestion.
Question 1. Lines 42-43: the authors should define what they means with good piezoresistive effect. Graphene-based materials have shown gauge factors among the highest ever reported. The authors can use this reference: https://www.nature.com/articles/srep00870.
Answer: We thank the reviewer for the valuable comments. We have added the following text in the revised manuscript and cited the recommended reference.
Line 43-Line 44
“Graphene-based materials have shown the highest gauge factors among ever reported literatures and can be transferred to various flexible substrates [18].”
Question 2. Line 50: use the past tense when describing something that was already done. "They proved".
Answer: We extend our gratitude to the reviewer for pointing out the issue. We have corrected it in the revised manuscript, as shown in Line 50.
Question 3. Line 57: This sentence is not clear.
Answer: Thanks for the valuable suggestion. We have corrected it in the revised manuscript. In case of misleading, we change ‘it is’ to ‘the method is’ as shown in Line 57.
Question 4. -line 135: the authors should report also the value of sheet resistance (Ohm/sq) of the material.
Answer: The value of sheet resistance of the material is ~3.25 Ohm/sq. A GNPs sheet comprises more GNPs, under pressing, these GNPs are overlapped or interleaved, which dramatically reduced the volume fraction of the gaps, hollow and cracks between the platelets, thus offering efficient conductive networks.
Question 5. When the authors discuss hysteresis (Figure 3d), it would be interesting to know how much larger is the value of the resistance after the release of the stress.
Answer: After the release of the stress, the value of the resistance is shown in the following figure.
Thanks for your valuable comments!
Sincerely yours,
Lisheng Xu
March 30, 2019

Reviewer 2 Report
The authors have addressed most of my comments. I think the manuscript is now greatly improved compared to the initial version. However, I have two more comments which I believe need to be addressed before the manuscript is accepted for publication.
1) Page 5, Figure 3:
The authors have tested the piezoresistivity of graphene sheets before and after sandwiched by the PU film. It is interesting that the stretchability has been greatly increased from around 10% to 100%.
One error is that the Figure 3a and Figure 3b’s caption is not consistent with figures, which should be swapped.
2) Measurement of finger bending action
The authors did not understand my question about the Y-axis of Figures 5-7. The authors present all the data in the form of relative resistance change R-R0/R0 in Figure 3, which is the most important figure in the paper. Therefore, to be consistent and easy to compare and understand, the data here in figures 5-7 should also present in the form of relative resistance change R-R0/R0. The voltage measured should be converted to resistance.
Author Response
Dear reviewer, We thank the referee for carefully reading this manuscript and providing useful comments and suggestions. We have modified this manuscript to accommodate all the comments as indicated below (comments are indicated in bold; our responses are provided in italics and blue). All the specific changes have been made in response to the points in the revised manuscript in red. Comments: The authors have addressed most of my comments. I think the manuscript is now greatly improved compared to the initial version. However, I have two more comments which I believe need to be addressed before the manuscript is accepted for publication. Answer: We gratefully thank the reviewer for the positive comments. Question 1. Page 5, Figure 3: The authors have tested the piezoresistivity of graphene sheets before and after sandwiched by the PU film. It is interesting that the stretchability has been greatly increased from around 10% to 100%. One error is that the Figure 3a and Figure 3b’s caption is not consistent with figures, which should be swapped. Answer: We thank the reviewer for pointing out the issue. We have corrected the mistake in the revised manuscript. Question 2. Measurement of finger bending action The authors did not understand my question about the Y-axis of Figures 5-7. The authors present all the data in the form of relative resistance change R-R0/R0 in Figure 3, which is the most important figure in the paper. Therefore, to be consistent and easy to compare and understand, the data here in figures 5-7 should also present in the form of relative resistance change R-R0/R0. The voltage measured should be converted to resistance. Answer: Once again, we extend our gratitude to the reviewer for the comments. After repeated experiments and studies, we find that it is not scientific to calculate the input signals by the output signals in the filter circuit. It is unscientific to convert the attenuated voltage value directly to (R-R0)/R0. What is more, the physiological signals belong to weak signals, which usually are disturbed by the power frequency interference and baseline drift. The noise or interference can be either greatly reduced or even eliminated by using signal conditioning or filtering techniques. If wearable sensors were commercial, a signal processing circuit is needed for practical application. Hence, we have developed a signal processing circuit to output the detection signals. From the logic of the paper, it is better to use the final output signal in Figure 5-7. We thank the referee for providing useful comments and suggestions again. Sincerely yours, Lisheng Xu March 30, 2019
